RESEARCH COMMUNICATION

# Global warming reduces leaf-out and flowering synchrony among individuals

**Constantin M Zohner[1]\*, Lidong Mo[1], Susanne S Renner[2]**

[1]Institute of Integrative Biology, ETH Zurich (Swiss Federal Institute of Technology), Zurich, Switzerland; [2]Department of Biology, Systematic Botany and Mycology, University of Munich (LMU), Munich, Germany

**Abstract** The temporal overlap of phenological stages, phenological synchrony, crucially influences ecosystem functioning. For flowering, among-individual synchrony influences gene flow. For leaf-out, it affects interactions with herbivores and competing plants. If individuals differ in their reaction to the ongoing change in global climate, this should affect population-level synchrony. Here, we use climate-manipulation experiments, Pan-European long-term (>15 years) observations, and common garden monitoring data on up to 72 woody and herbaceous species to study the effects of increasing temperatures on the extent of leaf-out and flowering synchrony within populations. Warmer temperatures reduce in situ leaf-out and flowering synchrony by up to 55%, and experiments on European beech provide a mechanism for how individual differences in day-length and/or chilling sensitivity may explain this finding. The rapid loss of reproductive and vegetative synchrony in European plants predicts changes in their gene flow and trophic interactions, but community-wide consequences remain largely unknown.

**Editorial note:** This article has been through an editorial process in which the authors decide how to respond to the issues raised during peer review. The Reviewing Editor's assessment is that all the issues have been addressed (see decision letter).

DOI: https://doi.org/10.7554/eLife.40214.001

\*For correspondence:
constantin.zohner@t-online.de

**Competing interests:** The authors declare that no competing interests exist.

## Introduction

The structure and functioning of ecosystems crucially depends on the timing of annually repeated life stages, such as leaf-out and flowering (*Ims, 1990*; *Fitter and Fitter, 2002*; *Sherry et al., 2007*; *Thackeray et al., 2016*). Anthropogenic climate warming is causing advanced leaf-out and flowering in both herbs and trees, and this is affecting growth and reproductive success (*Menzel and Fabian, 1999*; *Chuine and Beaubien, 2001*; *Elzinga et al., 2007*; *Chuine, 2010*). Warmer springs and summers are also causing leaf-out and flowering to spread out over longer periods because the sensitivity to changing abiotic conditions differs among species (*Fitter and Fitter, 2002*; *Sherry et al., 2007*; *Zohner et al., 2017*; *Laube et al., 2014*; *Zohner et al., 2016*). Leaf-out and flowering times might also spread out within species (*CaraDonna et al., 2014*), potentially reducing phenological synchrony among individuals. For leaf-out, inter-individual synchrony affects interactions with foliovores and competing plants (*Hart et al., 2016*). For flowering, reduced inter-individual synchrony should adversely affect gene flow by reducing cross-pollination and fruit set (*Augspurger, 1981*) and alter co-flowering patterns within communities (*CaraDonna et al., 2014*; *Forrest et al., 2010*). To detect such possible effects of climate warming on within-population synchrony, a range of herbs and trees, representing different leaf-out and flowering strategies, needs to be studied.

Here, we use a combination of climate-manipulation experiments, common-garden monitoring, and long-term Central European in situ observations to analyze effects of warming on intraspecific phenological synchrony. The long-term data were obtained from the Pan European Phenology Project (http://www.pep725.eu, hereafter PEP) and consisted of 12,536 individual time series (each

minimally 15 years long), comprising the leaf-out times of nine dominant tree species and the flowering times of six tree species, four shrubs, and five herbs (see Materials and methods and the distribution of the sites in *Figure 1a*, *Figure 1—figure supplements 1* and *2*).

## Results and discussion

To analyze the PEP data, the study area was divided into pixels of one-degree resolution (~110×85 km), and leaf-out synchrony (LOS) and flowering synchrony (FLS) in a given year were then calculated as the standard deviation of leaf-out or flowering date for all individuals within a pixel (note that the data were cleaned to ensure that observed individuals were the same between years; see Materials and methods). For each pixel and each phenological stage (leaf-out or flowering), we determined preseason as the period 60 days before the average leaf unfolding or flowering date within the respective pixel.

As expected, within pixels, species' mean leaf-out dates were negatively correlated with preseason temperature (98% of observation series statistically significant at $p<0.05$), with a mean linear correlation coefficient of $-0.76 \pm 0.03$ (mean ± 95% confidence interval), predicting an average advance of $4.3 \pm 0.2$ days per each degree warming. Similarly, in more than 99% of pixels, the mean flowering dates were negatively correlated with the preseason temperature (91% statistically significant at $p<0.05$), with a mean linear correlation coefficient of $-0.75 \pm 0.10$, predicting an average advance of $4.6 \pm 0.2$ days per each degree warming.

Higher preseason temperatures had a negative effect on LOS in eight of the nine species (*Figure 1c*, *Figure 1—figure supplement 1*) and on FLS in 10 out of 15 species (*Figure 1d*, *Figure 1—figure supplement 2*). None of the species exhibited a positive effect. Across all species, preseason temperature negatively affected LOS in 78% of analyzed pixels (15% statistically significant at $p<0.05$), that is, the standard deviation of inter-individual leaf-out times increased by $0.45 \pm 0.07$ (mean ± CI) days per degree of warming, with a mean linear correlation coefficient of $0.19 \pm 0.03$. Significant positive effects of preseason temperature on LOS appeared in fewer than 1% of pixels. The species showing the strongest decline in LOS related to warmer preseason temperatures was European beech (*Fagus sylvatica*; *Figure 1a*): preseason temperature negatively affected LOS in 95% of analyzed pixels (39% statistically significant), with the standard deviation of inter-individual leaf-out times increasing by $0.61 \pm 0.05$ days per degree of warming (*Figure 1b*). When modelling the distribution of leaf-out dates within pixels, we found that preseason warming increases the inter-individual variation in leaf-out times by up to 55%, which equates to lengthening the period during which 95% of individuals in a pixel leaf-out by 11 days (*Figure 1e* and *Figure 1—figure supplement 3*).

Across all species, preseason temperature negatively affected FLS in 75% of analyzed pixels (18% statistically significant), with the standard deviation of inter-individual flowering times increasing by $0.35 \pm 0.15$ days per degree of warming and a mean linear correlation coefficient of $0.15 \pm 0.06$ (*Figure 1d* and *Figure 1—figure supplement 2a*). A significant positive effect of preseason temperature on FLS was found in only 2% of pixels. The species showing the strongest decline in FLS related to warmer preseason temperatures was the European alder (*Alnus glutinosa*): preseason temperature negatively affected FLS in 91% of analyzed pixels (33% statistically significant), with the standard deviation of inter-individual flowering times increasing by $0.91 \pm 0.27$ days per degree of warming. When modelling the distribution of flowering dates within pixels, we found that preseason warming increases leaf-out variation by up to 51%, which equates to lengthening the period during which 95% of individuals in a pixel initiate flowering by 23 days (*Figure 1f* and *Figure 1—figure supplement 4*). In species, such as the crocus *Colchicum autumnale* and the heath *Calluna vulgaris*, where preseason temperature had little effect on the mean flowering date, preseason temperature also had little effect on FLS (*Figure 1—figure supplements 4* and *5*).

To cross-validate the results obtained from the PEP data, we used common garden data consisting of leaf-out information on 209 individuals in 59 temperate woody species (minimally three individuals per species) observed in the Munich Botanical garden from 2013 to 2018. A Bayesian hierarchical model, including preseason temperature as predictor variable, the standard deviation of inter-individual leaf-out times per year as response variable, and species as a random effect, showed a significantly negative effect of preseason temperature on LOS (lower panel *Figure 2a*). On

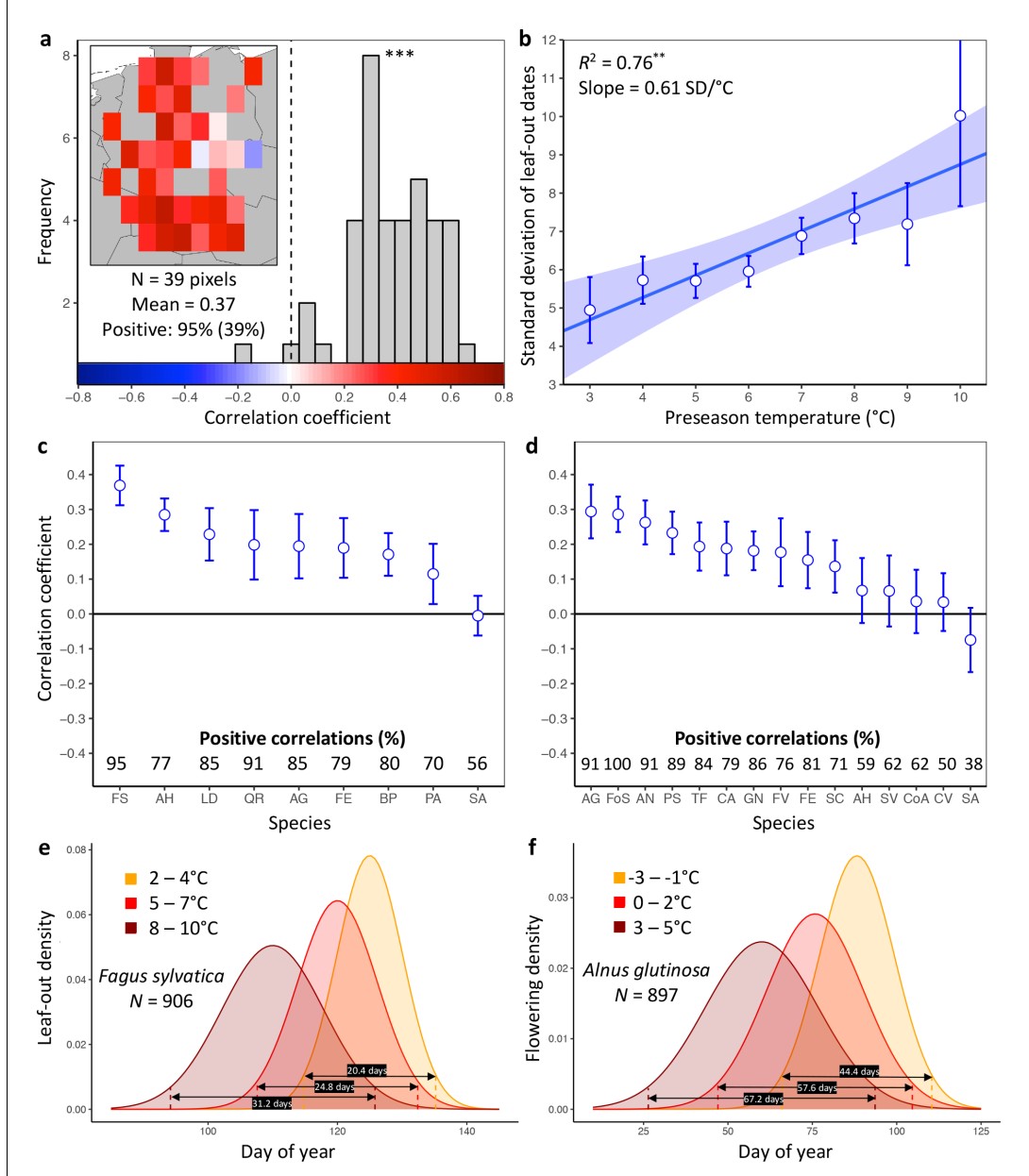

**Figure 1.** Loss of inter-individual synchrony in leaf-out and flowering with increasing temperatures. (a) Frequency distribution showing the correlations between the standard deviation of inter-individual leaf-out times and spring temperature for *Fagus sylvatica* at 39 pixels (1° x 1° areas). Mean = Mean correlation coefficients across all sites (N) Positive = percentage of positive correlations and the percentage of statistically significant positive correlations (in parentheses). Inset shows a heat map of the correlations at the 39 pixels. (b) Effect of preseason temperature on the standard deviation of inter-individual leaf-out times (mean ± SEM) in *F. sylvatica* averaged across all years and sites. (c) (d) Mean Pearson correlation coefficients (± 95% confidence intervals) for the effect of spring temperature on the standard deviation of inter- individual leaf-out (c) or flowering times (d). *Positive correlations* = percentage of the total number of positive correlations. See *Figure 1—figure supplements 1b* and *2b* for number of sites (1° x 1° areas) in which the relationship was analyzed. (e) (f) Distributions of inter-individual (e) leaf-out dates in *F. sylvatica* and (f) flowering dates in *Alnus glutinosa* under different spring temperatures. N = Number of available year x pixel (1° x 1° areas) combinations. To model the distributions (means and standard deviations), mixed-effects models were applied including site (pixel) as a random effect. See for distributions of all 20 analyzed species. AG, *Alnus glutinosa*; AH, *Aesculus hippocastanum*; AN, *Anemone nemorosa*; BP, *Betula pendula*; CA, *Corylus avellana*; CoA; *Colchicum autumnale*; CV, *Calluna vulgaris*; FE, *Fraxinus excelsior*; FoS, *Forsythia suspensa*; FS, *Fagus sylvatica*; FV, *Fragaria vesca*; GN, *Galanthus nivalis*; LD, *Larix decidua*; PA, *Picea abies*; PS, *Prunus spinosa*; QR, *Quercus robur*; SA, *Sorbus aucuparia*; SC, *Salix caprea*; SV, *Syringa vulgaris*; TF, *Tussilago farfara*.
DOI: https://doi.org/10.7554/eLife.40214.002

The following figure supplements are available for figure 1:

*Figure 1 continued on next page*

*Figure 1 continued*

**Figure supplement 1.** Effects of preseason temperature on inter-individual leaf-out synchrony (LOS), using PEP data.

DOI: https://doi.org/10.7554/eLife.40214.003

**Figure supplement 2.** Effects of preseason temperature on inter-individual flowering synchrony (FLS), using PEP data.

DOI: https://doi.org/10.7554/eLife.40214.004

**Figure supplement 3.** The effect of preseason temperature on inter-individual (within-population) leaf-out distributions.

DOI: https://doi.org/10.7554/eLife.40214.005

**Figure supplement 4.** The effect of preseason temperature on inter-individual (within population) flowering distributions.

DOI: https://doi.org/10.7554/eLife.40214.006

**Figure supplement 5.** In species in which preseason temperature has little effect on the mean flowering date, preseason temperature also has little effect on FLS.

DOI: https://doi.org/10.7554/eLife.40214.007

average, across all 59 species, the standard deviation of inter-individual leaf-out times increased by 0.26 ± 0.10 (mean ± CI) days per degree of warming.

Which factors cause the loss of inter-individual synchrony under climate warming? One possibility is that individuals reach their forcing sums (accumulated warming required for leaf-out or flowering) over a longer period because 'within-spring warming speed' may be decreasing, flattening the temperature curve during spring (*Wang et al., 2015*; *Wang et al., 2016*). Thus, while the time span among individual leaf-out times might increase, differences in the forcing sums required until leaf-out or flowering among individuals might remain similar (*Figure 2—figure supplement 2a*). To test this, we additionally calculated leaf-out/flowering synchrony as the standard deviation in individual forcing requirements (degree-days [DD] from 1 January until leaf-out/flowering) [hereafter referred to as LOS-DD and FLS-DD] for both the PEP and Munich common garden data. In both data sets, we found a strong (albeit slightly weaker compared to the LOS/FLS analysis) negative relationship between preseason temperature and LOS-DD, that is, individual differences in the forcing sums required until leaf-out or flowering are increasing with warmer preseasons (*Figure 2a* and *Figure 2—figure supplement 1*). We also simulated synchrony of spring phenology based on the Munich Jan–May temperatures over the past 60 years, assuming that phenology is solely driven by degree-day accumulation (no effect of day length or winter chilling; see *Figure 2—figure supplement 2b*). This simulation revealed losses of synchrony under warmer preseasons (regression coefficients between 0.15 and 0.43 SD/°C; *Figure 2—figure supplement 2c*), but those simulated losses are small relative to the actual losses inferred from in situ observations (see red arrows in *Figure 1—figure supplement 1a* and *Figure 1—figure supplement 2a*). Together, those results show that a flattening temperature curve during spring is not sufficient to explain the declining inter-individual synchrony in the 72 species analyzed here.

Warmer preseasons in spring are associated with both reduced accumulation of winter chilling and shorter day-lengths at spring onset, and previous experiments on plant phenological strategies have shown pronounced differences among species in their reactions to day length and winter chilling (*Zohner et al., 2017*; *Laube et al., 2014*; *Zohner et al., 2016*). To test whether similar differences within species might explain the decrease in LOS and FLS under climate warming detected in our in-situ data, we designed experiments in which we exposed trees to different regimes of spring warming, winter chilling, and day length. We additionally tested for the relative effects of winter chilling and day length on LOS and FLS using the PEP and Munich common garden data (for each year and individual, we calculated winter chilling experienced until leaf-out and day length for the date when an individual's average forcing requirement had been reached).

A first experiment addressed inter-individual variation in spring warming ('forcing'), day length, and winter chilling requirements in 11 mature *Fagus sylvatica* trees growing in the vicinity of the botanical garden in Munich. Twigs were cut at three dormancy stages during winter and exposed to different day-length regimes (8 hr, 12 hr, or 16 hr light per day) and ambient spring-forcing conditions (mean daily temperature of 16°C). Note that in beech, leaf-out and flowering occur simultaneously because leaves and flowers are located on the same preformed shoots within overwintering buds. The results showed large differences in forcing and day-length requirements among individuals (*Figure 3a and b*): for example, while in individual 1, day length had no effect on the amount of warming required until leaf-out, in individual 11, warming requirements were >2 x lower under long-

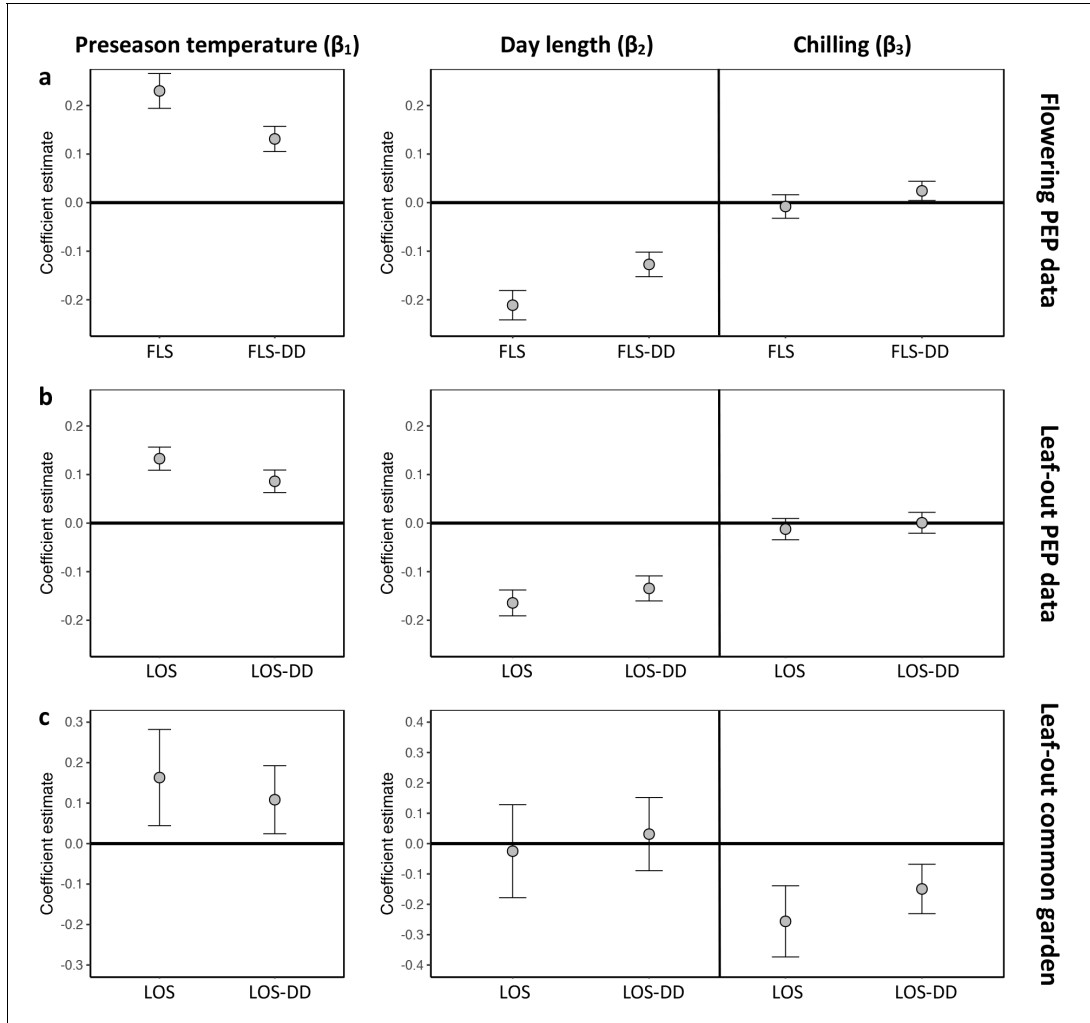

**Figure 2.** Hierarchical Bayesian models to test for the environmental drivers of inter- individual flowering (**a**) and leaf-out (**b, c**) synchrony. Plots show coefficient values (β) [means and 95% credible intervals] for *equations 6 and 7* (see Materials and methods). PEP data (**a, b**) or common-garden observations (**c**) were used for analysis. Left panels: The effect of preseason temperature on inter-individual phenological synchrony measured either as the standard deviation in leaf-out/flowering dates (LOS/FLS) or the standard deviation in degree-day (DD) requirements among individuals (LOS-DD/ FLS DD). Right panels: The effects of day length and winter chilling on inter-individual leaf-out synchrony. To account for within-species rather than among-species synchrony, all models include species random effects. The models using the PEP data (**a and b**) additionally include site random effects (1° pixels) to address within-pixel phenological synchrony. All variables were standardized to allow for direct effect size comparisons. $N$ = 13 (**a**), 9 (**b**), and 59 species (**c**).

DOI: https://doi.org/10.7554/eLife.40214.008

The following figure supplements are available for figure 2:

**Figure supplement 1.** Greater variation of forcing requirements among individuals with increasing preseason temperatures.

DOI: https://doi.org/10.7554/eLife.40214.009

**Figure supplement 2.** Does decreased LOS and FLS under climate warming result from a decrease in within-spring warming speed?

DOI: https://doi.org/10.7554/eLife.40214.010

day than under short-day conditions (*Figure 3b*). Chilling requirements differed little among individuals (compare slopes in *Figure 3c*).

In a second experiment, we cut twigs of the same 11 beech trees at eight dormancy stages during winter and exposed them to natural day length. Temperatures were the same as in experiment 1, that is, ambient. This allowed us to determine (i) the extent to which differential reliance on forcing, day length, and winter chilling (as inferred from experiment 1) explains LOS/FLS under natural light conditions, and (ii) the effect of warmer winter and spring conditions on LOS/FLS. Leaf-out

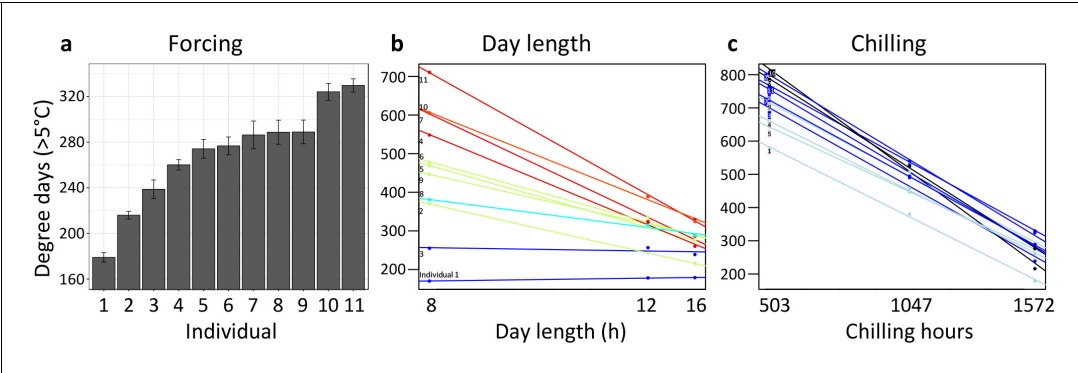

**Figure 3.** Individual differences in the forcing (**a**), day-length (**b**), and chilling (**c**) requirements among 11 beech trees (*F. sylvatica*; Experiment 1). (**a**) Mean ( ± SEM) forcing requirements (accumulative degree-days >5°C) until leaf-out under long chilling and constant 16 h day length. (**b**) Degree-days until leaf-out at 8 hr, 12 hr, and 16 h day length (collection date: 21 March 2015). Colours according to slope (red: steep slope; blue: no slope). (**c**) Degree-days until leaf-out under short, intermediate, and long chilling (collection dates: 22 Dec 2014, 6 Feb 2015, 21 March 2015) and 16 h day length. Colours according to slope (dark blue: steep slope; light blue: no slope).

DOI: https://doi.org/10.7554/eLife.40214.011

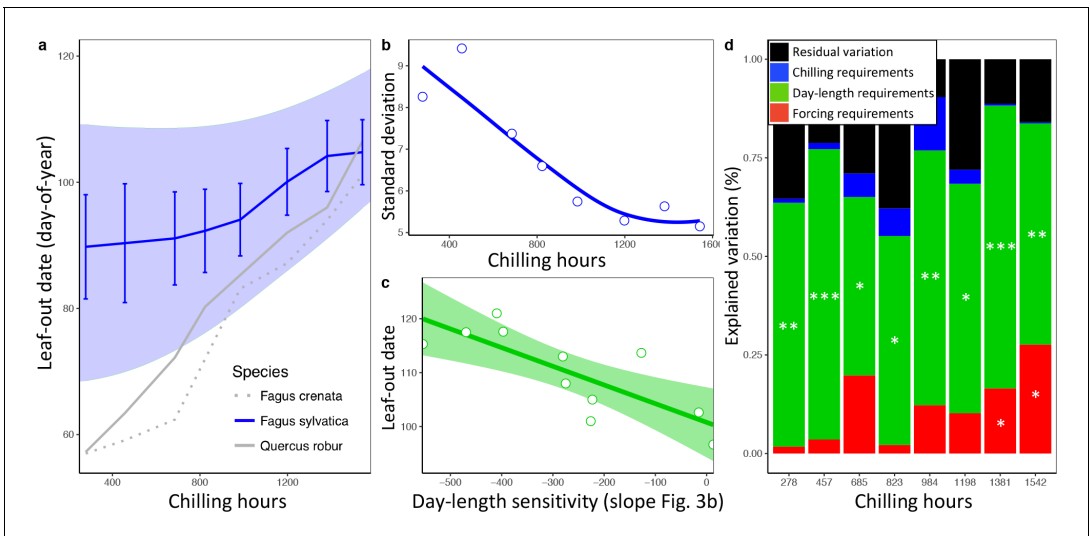

**Figure 4.** Loss of phenological synchrony with climate warming is explained by contrasting day-length sensitivities in *Fagus sylvatica*.**a**, **b**, Experiment 2. (**a**) Leaf-out dates of *Fagus sylvatica* (blue), *Fagus crenata* (dotted grey), and *Quercus robur* (grey) under varying winter lengths (chilling hours = sum of hours from 1 November until leaf-out with an average temperature between 0°C and 5°C). Bars show the standard deviation of average leaf-out dates among 11 *F. sylvatica* individuals. The shaded area shows the difference between the leaf-out date of the first flushing twig of the first individual and the last twig of the last individual to leaf- out, using a LOESS smoothing function. For *F. crenata* and *Q. robur*, we investigated one individual each and therefore do not report inter-individual variation. (**b**) Standard deviation of leaf-out dates among 11 *F. sylvatica* individuals at different winter lengths (chilling levels) and natural day length. (**c**) The effect of individual day-length sensitivity on the timing of leaf unfolding when twigs were collected on 10 December 2015. Note the reversed x-axis scale, that is, smaller values indicate higher day-length sensitivity. (**d**) Variables explaining the sequence of leaf-out dates of 11 *F. sylvatica* individuals at eight different chilling levels. The percentage of leaf-out variation (derived from the ANOVA sums of squares) that can be explained by individual forcing requirements (red), day-length requirements (green), chilling requirements (blue), and the remaining residuals, that is, unexplained variation (black). *p<0.05; **p<0.01; ***p<0.001.

DOI: https://doi.org/10.7554/eLife.40214.012

The following figure supplement is available for figure 4:

**Figure supplement 1.** Percent variation in leaf-out dates attributable to treatment effects and between- and within-individual variation within treatments.

DOI: https://doi.org/10.7554/eLife.40214.013

variation among individuals explained most of the total variation among twigs, with 52% attributable to between-individual variation, 33% to treatments, and only 15% to within-individual variation (*Figure 4—figure supplement 1*). As in the *in situ* data from the Pan European Phenology network, synchrony strongly decreased under warmer spring conditions (*Figure 4a,b*). We hypothesized that differences in day-length sensitivity among individuals (as documented for *F. sylvatica*; *Figure 3b*) can explain this: Under cold winter conditions, days are already long when spring warming occurs, reducing the effect of a tree's day length sensitivity on its leaf-out time, whereas with early spring warming, days are still short, preventing day-length sensitive trees from flushing. In natural populations, leaf-out advancement in day length-sensitive individuals, but not in day length-insensitive individuals, will thus increase the period of leaf-out under short day conditions. Both the experimental and the PEP in situ data confirm this idea, showing that (i) phenological synchrony among individuals strongly decreases under short day conditions (*Figures 2b* and *3b*) and (ii) differences in day-length requirements are the single most important factor explaining individual variation in leaf-out times (*Figure 4c,d*).

This insight explains why, especially in *Fagus sylvatica*, in which day length has the most pronounced effect on spring phenology (*Laube et al., 2014*; *Zohner et al., 2016*), LOS is strongly affected by preseason temperatures (*Figure 1c*). By contrast, in day-length insensitive species, such as silver birch *Betula pendula* and Norway spruce *Picea abies* (*Zohner et al., 2016*), preseason warming has a smaller (but still significant) effect on LOS, suggesting that heritable differences in day-length sensitivity are a major driver of phenological variation among individuals. In our common garden data, the standard deviation of inter-individual leaf-out times increased by $0.09 \pm 0.02$ (mean $\pm$ CI) days per decrease in one chilling day, and the standard deviation of inter-individual forcing requirements increased by $0.23 \pm 0.06$ degree-days per decrease in one chilling day (lower panel *Figure 2b*), indicating that individual differences in the sensitivity to winter chilling also contribute to the observed loss of phenological synchrony under climate warming.

What biological consequences can be expected from less synchronized leaf-out and flowering of the individuals of a species? With regard to vegetative development, precocious leaf unfolding under warm springs increases the risk of late frost damage (*Augspurger, 2013*; *Kollas et al., 2014*; *Vitasse et al., 2014*), but also potential carbon gain due to earlier photosynthetic activity (*Keenan et al., 2014*). This risk-return trade-off will affect selection on suitable genotypes under future conditions, and the increasing spread of leaf-out should increase the selective importance of spring phenology. Whether opportunistic phenological strategies (relying on temperature as the main trigger) or conservative strategies (relying on day length and/or winter chilling as a buffer against highly variable spring temperatures) will be favored in the future will be region-specific, depending on the relative advancement rates of spring warming and late frost events. In continental regions, where the advent of spring is relatively invariable (low late frost risk), phenological strategies reliant on temperature should be favored (*Zohner et al., 2017*; *Körner and Basler, 2010*).

With regard to flowering, decreased synchrony among individuals, as already strongly evident in *Alnus glutinosa* (*Figure 1f*), should lead to reduced inter-individual pollen transfer. Strong divergence in flowering times among individuals also might lead to assortative mating (depending on incompatibility systems), possibly promoting local adaptation (*Antonovics and Bradshaw, 1970*; *Kirkpatrick, 2000*; *Weis and Kossler, 2004*) and should act as a buffer against climate change-induced phenological mismatch between plants and leaf-feeding or pollen-collecting insects (*Renner and Zohner, 2018*). Rapid adaptive responses, for instance a filtering out of extreme phenotypes through increased mortality or reduced reproduction, might counteract warming-induced losses of inter-individual synchrony. Such selection of the standing variation can occur very rapidly, at least in herbaceous plants (*Jump and Penuelas, 2005*; *Fakheran et al., 2010*).

While our results show that climate warming causes a loss of phenological synchrony among the individuals of a population, a study of leaf-out along elevational gradients in four European tree species, between 1960 – 2016, revealed that leaf-out times at higher and lower elevations are today compressed into a shorter time window compared to 58 years ago (*Vitasse et al., 2018*). These findings do not contradict those of the present study because populations growing at high elevations were able to advance their phenology more than those at lower elevations for which chilling and/or day-length requirements are no longer fulfilled (*Figure 5*). As a result, the leaf-out times of high- and low-elevation populations are converging (*Vitasse et al., 2018*). At the same time, however, differences in day-length sensitivity (as well as chilling and temperature sensitivity) among the individuals

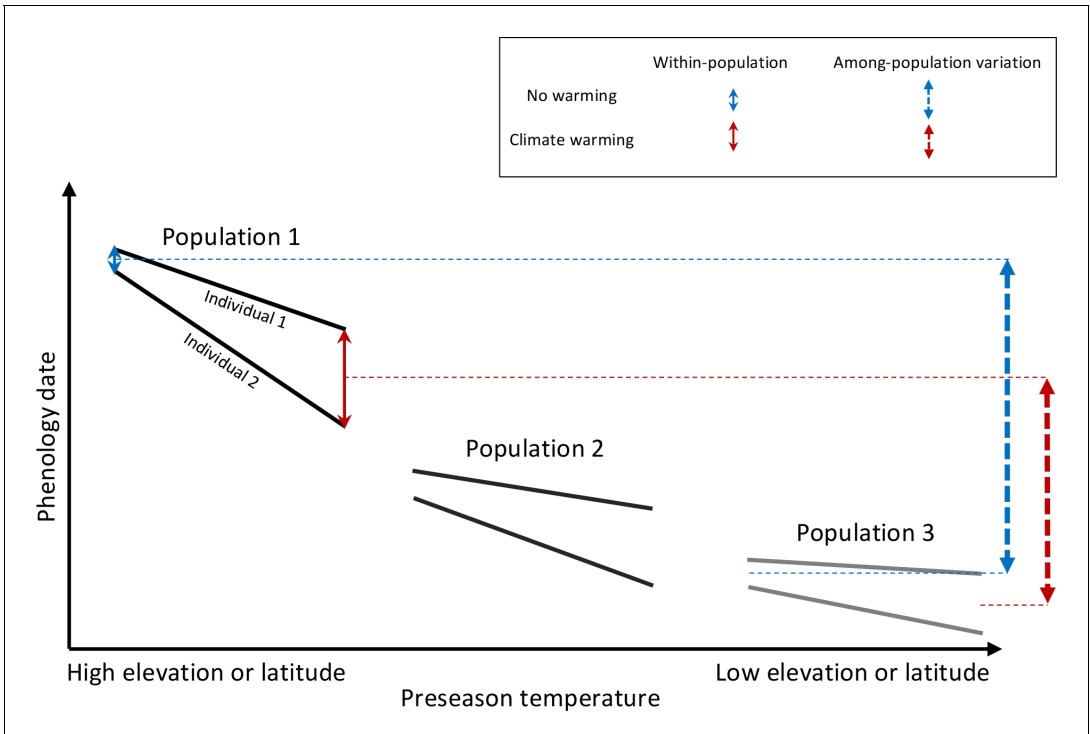

**Figure 5.** Schematic representation of within- and among-population phenological synchrony in response to climate warming. As demonstrated in this study, inter-individual synchrony within a population will decrease under warmer preseason temperatures because individuals differ in their sensitivity to temperature. Within-population variation under ambient or warmed preseason temperatures is illustrated by the solid blue and red arrows, respectively. By contrast, phenological synchrony among populations is expected to increase, given that populations in warm regions (Population 3) will advance their phenology less than populations in cold regions (Population 1). This is illustrated by the dashed blue and red arrows, showing that the difference in the average phenological date between Population 1 and 3 is smaller under warmer preseasons (red dashed arrow) than under ambient preseason temperatures (blue dashed arrow).

DOI: https://doi.org/10.7554/eLife.40214.014

at any one elevation under climate warming are resulting in diverging flowering and leaf-out times within populations. A likely side-effect of increasing phenological variation within populations is a community-level reduction of variation between species, which could reduce phenological niche differentiation between species and alter competitive environments (*CaraDonna et al., 2014*).

The overall prediction from the present findings is that human-caused climate warming is leading to plant phenologies that are more heterogeneous within populations and more uniform among populations (over altitude or latitude) and among species (within communities). The rapid loss of reproductive and vegetative synchrony in European plant populations also predicts changes in their gene flow and trophic interactions, although community-wide consequences are presently unknown.

## Conclusion

The synchrony of developmental stages among organisms is a critical aspect of ecosystem functioning. Here, based on ground observations and climate-manipulation experiments, we show that global warming is altering within-population synchrony of leaf-out and flowering dates in temperate plants, with warmer temperatures reducing inter-individual synchrony by up to 55%. Experiments suggest that individual differences in the sensitivity to day-length and/or winter chilling underlie the loss of synchrony, and future climate warming is expected to further strengthen this trend. These results predict consequences for gene flow and trophic interactions, but also emphasize the importance of adaptation when forecasting future plant growth and productivity.

## Materials and methods

### Analysis of leaf-out and flowering synchrony (LOS and FLS) using the PEP database

Data sets

In situ phenological observations were obtained from the Pan European Phenology network (*Templ et al., 2018*), which provides open-access European phenological data. Leaf-out dates were analyzed for nine species, flowering dates for 15. Data from Germany, Austria, and Switzerland were used for the analysis. For the angiosperm woody species, leaf-out was defined as the date when unfolded leaves, pushed out all the way to the petiole, were visible on the respective individual (BBCH 11, Biologische Bundesanstalt, Bundessortenamt und Chemische Industrie). For the two conifers *Larix decidua* and *Picea abies,* leaf-out was defined as the date when the first needles started to separate ('mouse-ear stage'; BBCH 10). Flowering was defined as the date of beginning of flowering (BBCH 60). We removed (i) individual time series, for which the standard deviation of phenological observations across years was higher than 25 and (ii) leaf-out and flowering dates that deviated from an individual's median more than three times the median absolute deviation (moderately conservative threshold) (*Vitasse et al., 2018*). Here, we consider time series as being equal to individuals because, in the PEP database, the same individual is usually observed for many years. However, we acknowledge that this does not necessarily have to be the case for all time series, as individuals might die or not be detectable due to other reasons.

Analysis

To test for an effect of spring temperature on inter-individual leaf-out synchrony (LOS) and flowering synchrony (FLS), we divided the study area into pixels of one degree resolution (~110×85 km), an area that can reasonably be considered as reflecting populations, at least for wind-pollinated woody species (see discussion on herbs in the main text). To allow for within-pixel comparisons of LOS and FLS between years, data from the same individuals had to be used each year. To achieve this, we kept only pixels for which there were at least three individuals with data for the same 15 years. For each pixel, we deleted all (i) individuals growing at altitudes that deviated by >200 m from the average altitude of all individuals within the pixel, and (ii) years that had less than 90% plant-coverage, that is, data from at least 90% of the individuals within the pixel had to be available for the respective year, otherwise the year was excluded from the analysis. This data cleaning left us with a total of 12,536 individuals, 317,672 phenological observations (individuals x year), and a median time-series length of 25 years (minimally 15 years, maximally 48 years). The number of individuals within pixels (per species and phenological stage) ranged between 3 and 53 (median = 12). See *Figure 1—figure supplement 1b* and *Figure 1—figure supplement 2b* for information on the number of pixels used per species.

For each year and species, LOS and FLS within pixels were then calculated as the standard deviation of leaf-out or flowering dates. Additionally, we calculated the standard deviation of forcing requirements among individuals (subsequently referred to as LOS-DD [leaf-out synchrony degree-days] and FLS-DD [flowering synchrony degree-days]) to test if greater phenological variation among individuals can be explained by increasing variation in forcing requirements. Individual forcing requirements until leaf-out were calculated as the sum of degree-days (DD) from 1 January until leaf-out or flowering using 5°C as base temperature (e.g., *Zohner and Renner, 2015*):

$$DD_{sum}(t) = \sum_{t_0}^{t_{LO}} T_t - 5$$

where $DD_{sum}$ is the accumulated degree days until leaf unfolding, $t_{LO}$ is the day of leaf unfolding, $T_t$ is the mean daily temperature on day $t$, and $t_0$ is the start date for forcing accumulation, which was fixed at 1 January. For each year and species, LOS-DD and FLS-DD within pixels were then calculated as the standard deviation of forcing requirements until leaf-out or flowering dates.

The daily mean air temperature at each site was derived from a gridded climatic data set of daily mean temperature at 0.5° spatial resolution (approximately 50 km, ERA-WATCH) (*Beer et al., 2014*). For each year, preseason temperature within pixels was defined as the average temperature during the 60 days prior to the average leaf unfolding or flowering date within the respective pixel, which is

the period for which the correlation coefficient between phenological event and temperature is highest (*Fu et al., 2015*).

To test if shortened day lengths or reduced winter chilling explain the decrease in phenological synchrony under warmer preseasons, for each year, pixel, and species, we calculated the average chilling hours until leaf-out or flowering and the average day length (DL) at the date (DOY) when the average forcing requirements until leaf-out or flowering were fulfilled. Chilling hours were calculated on basis of 6-hourly temperature data (CRU-NCEP, spatial resolution of 0.5°; https://crudata.uea.ac.uk/cru/data/ncep/), as the sum of hours from 1 November until leaf-out/flowering with an average temperature between 0°C and 5°C (e.g., *Vitasse et al., 2018*):

$$Ch_{sum}(t) = \sum_{to}^{t_{LO}} 1 \quad if \ 0 \leq T_t \leq 5 \tag{1}$$

where $Ch_{sum}$ is the sum of chilling hours until leaf unfolding, $t_{LO}$ is the day of leaf unfolding, $T_t$ is the hourly mean temperature on hour $t$, and $t_0$ is the start date for chilling accumulation, which was fixed at 1 November in the year before leaf unfolding.

DL was calculated as a function of latitude and DOY (*Forsythe et al., 1995*):

$$DL = 24 - \frac{24}{\pi} cos^{-1} \left[ \frac{sin\frac{0.8333\pi}{180} + sin\frac{L\pi}{180} sin \varphi}{cos\frac{L\pi}{180} * cos\varphi} \right] \tag{2}$$

$$\varphi = sin^{-1}(0.29795 * cos\theta) \tag{3}$$

$$\theta = 0.2163108 + 2 * tan^{-1}(0.9671396 * \tan(0.0086 * (DOY - 186))) \tag{4}$$

where $L$ is the latitude of the phenological site.

## Statistical analyses

Within each pixel we applied linear models to test for an effect of preseason temperature, day length, and winter chilling on phenological synchrony (LOS, LOS-DD, FLS and FLS-DD). We then determined the frequency distributions for the correlation coefficients between phenological synchrony and preseason temperature across all species and sites. For each species, we applied $t$-tests to detect whether the average of all correlation coefficients obtained for each pixel differs from zero. To model changes in the distribution of within-pixel leaf-out and flowering dates (means and standard deviations) in response to temperature, we applied mixed-effects models using average leaf-out/flowering dates or LOS/FLS as response variables, preseason temperature as explanatory variable, and site as a random effect to control for the use of different sites in the model.

Additionally, we applied a hierarchical Bayesian model to test for the relative effects of preseason temperature, winter chilling, or day-length on (i) inter-individual variation in leaf-out/flowering date (LOS / FLS) and (ii) inter-individual variation in forcing requirements until leaf-out/flowering (LOS-DD / FLS-DD). The use of a Bayesian framework allowed us to fit slope parameters across traits simultaneously without concerns of multiple testing or $P$-value correction. All models included random intercept effects for (i) species (to address within-species rather than between species phenological synchrony) and (ii) pixels (to address within-pixel rather than between-pixel phenological synchrony). Our model includes four dependent continuous variables (LOS, LOS-DD, FLS, and FLS-DD) that are normally distributed with mean μ, variance $\sigma^2$, and correlation structure Σ, hereafter referred to as *dependent*:

$$dependent_i \sim N(\mu_{dependent\ i}, \sigma^2, \Sigma) \tag{5}$$

Regression components of the model are of the form:

$$\mu_{dependent\ i} = \alpha_1 + \beta_1 \times preseason\ tmp_i + species_i + pixel_i \tag{6}$$

$$\mu_{dependent\ i} = \alpha_1 + \beta_2 \times daylength_i + \beta_3 \times chilling_i + species_i + pixel_i \tag{7}$$

where the term $\alpha$ refers to the intercept, $\beta$ to the estimated slopes of the respective variable in *Figure 2*, and *dependent* refers to synchrony values (*i*) [LOS, LOS-DD, FLS, or FLS-DD]. To allow for direct effect size comparisons, all continuous variables were standardized by subtracting their mean and dividing by 2 SD before analysis (*Gelman and Hill, 2007*). The resulting posterior distributions are a direct statement of the probability of our hypothesized relationships. Effective posterior means ± 95% confidence intervals are shown in *Figure 2*.

To parameterize our models, we used the JAGS implementation (*Plummer, 2003*) of Markov chain Monte Carlo methods in the R package R2JAGS (*Y-S and Yajima, 2014*). We ran three parallel MCMC chains for 200,000 iterations with a 50,000-iteration burn-in and evaluated model convergence with the Gelman and Rubin (*Gelman and Rubin, 1992*) statistic. Noninformative priors were specified for all parameter distributions, including normal priors for $\alpha$ and $\beta$ coefficients (fixed effects; mean = 0; variance = 1,000), and uniform priors between 0 and 100 for the variance of the random intercept effects, based on de Villemereuil and colleagues (*de Villemereuil et al., 2012*). All statistical analyses relied on R 3.2.2(*R Core Team, 2018*).

## Analysis of leaf-out synchrony (LOS) using common-garden data from 2013 to 2018

Between 2013 and 2018 we observed the leaf-out dates of 209 individuals in 59 temperate woody species (minimally three individuals per species) in the Munich Botanical garden (see Supplementary Materials *Supplementary file 1* for a list of species). An individual was scored as having leafed out when at least three branches had unfolded leaves pushed out all the way to the petiole (*International Phenological Gardens of Europe, 2017*). To test whether the trends observed in the PEP analysis are consistent with our common garden data, the same parameters (LOS, LOS-DD, preseason temperature, winter chilling, and day length) were calculated as described above (*Analysis of leaf-out and flowering synchrony (LOS and FLS) using the PEP database*). We then applied hierarchical Bayesian models including species random effects (see paragraph above) to test for the effects of preseason temperature, winter chilling, and day-length on LOS and LOS-DD.

## Twig cutting experiments and phenological scoring

To study the extent of intraspecific variation in leaf-out strategy (within-species variation in day length, chilling, and forcing requirements) and its implications under climate warming, we conducted twig-cutting experiments on mature *Fagus sylvatica* individuals growin in the vicinity of Munich. Experiments have demonstrated that twig cuttings precisely mirror the phenological behavior of their donor plants and therefore are adequate proxies for inferring phenological responses of adult trees to climatic changes (*Zohner and Renner, 2015*; *Vitasse and Basler, 2014*). We used twigs approximately 50 cm in length, and immediately after cutting, we disinfected the cut section with sodium hypochlorite solution (200 ppm active chlorine), cut the twigs a second time, and then placed them in 0.5 l glass bottles filled with 0.4 l cool tap water enriched with the broad-spectrum antibiotics gentamicin sulfate (40 µg l$^{-1}$; Sigma-Aldrich, Germany) (*Zohner et al., 2016*; *Zohner and Renner, 2015*). We then transferred the cut twigs to climate chambers and kept them under short (8 hr), intermediate (12 hr), or long day (16 hr) conditions (see Experiment one below), or natural day length (Experiment two below). Temperatures in the climate chambers were held at 12°C during the night and 20°C during the day, with an average daily temperature of 16°C to simulate forcing temperatures. Illuminance in the chambers was about eight klux (~100 µmol s$^{-1}$ m$^{-2}$). Relative air humidity was held between 40% and 60%. To account for within-individual variation, we used 10 replicate twigs per individual treatment and monitored bud development every second day. For each individual and treatment, we then calculated the mean leaf-out date out of the first eight twigs that leafed out. A twig was scored as having leafed out when three buds had unfolded leaves pushed out all the way to the petiole (*International Phenological Gardens of Europe, 2017*). Forcing requirements until leaf-out were calculated as the sum of degree-days [outside of and in climate chambers] from 10 December (1$^{st}$ collection date) until leaf-out using 5°C as base temperature (e.g., *Zohner and Renner, 2015*). Chilling hours were calculated as the sum of hours from 1 November until leaf-out with an average temperature between 0°C and 5°C.

## Experiment 1: Differences in day length sensitivity among *Fagus sylvatica* individuals

In winter 2014/2015, twigs of 11 individuals (10 replicate twigs per individual and treatment) of *Fagus sylvatica* were collected at three dates during winter (22 Dec 2014, 6 Feb 2015, and 21 Mar 2015) and brought into climate chambers. Additionally, we collected twigs from one individual each of *Fagus crenata* and *Quercus robur*. Temperatures in the chambers ranged from 12°C during night to 20°C during day, with an average daily temperature of 16°C. Day length in the chambers was set to 8 hr, 12 hr, or 16 hr.

Individual day length sensitivity was defined as the slope of the function between day-length treatment and accumulated degree days (>5°C) until leaf-out (twigs were collected on 21 March; see *Figure 3b*). The steeper the slope, the stronger the effect of day length on the amount of warming required for leaf-out. A flat slope indicates that day length has no effect on the timing of leaf-out.

Individual chilling sensitivity was defined as the slope of the function between chilling treatment (collection date) and accumulated degree days (>5°C) until leaf-out when twigs were kept under constant 16 h day length (see *Figure 3c*). The steeper the slope, the stronger the effect of chilling on the amount of warming required for leaf-out.

Individual forcing requirement was defined as the accumulated degree days (>5°C) until leaf-out under long chilling (21 March collection) and constant 16 h day length (see *Figure 3a*). Under such conditions, chilling requirements and day length requirements should be largely met, and thus the remaining variation in leaf-out dates should be largely attributable to differences in forcing (warming) requirements.

## Experiment 2: Different reactions to climate warming among *Fagus sylvatica* individuals

In winter 2015/2016, twigs from the same 11 individuals were harvested every two weeks (from 10 December until 21 March) and kept under the same temperature conditions applied in experiment 1 (12°C during night to 20°C during day), with natural day length. This allowed us to test if those individuals with no/little day length sensitivity would advance their leaf-out more under short winter conditions than day length-sensitive individuals, and to determine the relative effect of individual variation in day length requirements, chilling requirements and forcing requirements on leaf-out variation under different winter/spring conditions (*Figure 4*). Within-species leaf-out synchrony (LOS) was calculated as the standard deviation of individual leaf-out dates. To analyze which leaf-out cues (day length, chilling, and forcing requirements) best explain leaf-out variation among individuals, we applied a multivariate linear model, including individual forcing, day length, and chilling requirements (as inferred from experiment 1) as explanatory variables. To express the total variation in leaf-out dates that can be attributed to each trait, we used ANOVA sums of squares (see *Figure 4d*).

To infer which percentage of the variation in leaf-out dates is due to treatment effects, between-individual variation, or within-individual variation, we calculated variance components by applying a random-effects-only model including treatments and individuals as random effects (individuals nested within treatments) (*Figure 4—figure supplement 1*).

## Acknowledgements

We thank D März and V Sebald for help with the experiments and R Ricklefs for comments on the manuscript. This work benefited from the sharing of expertise within the DFG priority program SPP 1991 Taxon-Omics and support from DFG 603/25–1.

## Additional information

### Funding

No external funding was received for this work.

## Author contributions
Constantin M Zohner, Conceptualization, Formal analysis, Methodology, Writing—original draft, Writing—review and editing; Lidong Mo, Formal analysis; Susanne S Renner, Writing—review and editing

## Author ORCIDs
Constantin M Zohner  https://orcid.org/0000-0002-8302-4854
Susanne S Renner  https://orcid.org/0000-0003-3704-0703

## Decision letter and Author response
Decision letter https://doi.org/10.7554/eLife.40214.018
Author response https://doi.org/10.7554/eLife.40214.019

---

## Additional files

### Supplementary files
• Supplementary file 1. List of species.
DOI: https://doi.org/10.7554/eLife.40214.015
• Transparent reporting form
DOI: https://doi.org/10.7554/eLife.40214.016

### Data availability
All data generated during this study are included in the manuscript and supporting files. The in situ phenological data we analyzed can be found here following the creation of a free account: http://www.pep725.eu/data_download/data_selection.php

---

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
