## [Decision Letter]

[**Editorial note:** This article has been through an editorial process in which the authors decide how to respond to the issues raised during peer review. The Reviewing Editor's assessment is that all the issues have been addressed.]

Thank you for submitting your article "Loss of leaf-out and flowering synchrony under global warming" for consideration by *eLife*. Your article has been reviewed by three peer reviewers, including Bernhard Schmid as the Reviewing Editor and Reviewer #1, and the evaluation has been overseen by Ian Baldwin as the Senior Editor. The following individual involved in review of your submission has agreed to reveal their identity: Kentaro K Shimizu (Reviewer #2). A third reviewer remains anonymous.

The Reviewing Editor has highlighted the concerns that require revision and/or responses, and we have included the separate reviews below for your consideration. If you have any questions, please do not hesitate to contact us.

Summary:

This paper shows that increased pre-growing-season temperatures in central Europe lead to a wider spread of leaf-out and flowering-start days among individuals in several tree and other plant species. The potential causes – in particular within-population variation in preseason-temperature, photoperiod and chilling degree-day requirements – are analyzed. In addition, the slower increase in temperature in spring with increasing overall temperatures plays a role in the reduced synchrony of phenological events within species. The authors discuss potential consequences, i.e. reduced genetic exchange between individuals due to different flowering times and altered interactions with herbivores due to different leaf-out times.

Major concerns:

These are pointed out in the reviews below. In particular, please:

- discuss the issue of within- vs. between-species asynchrony/synchrony;

- justify the choice of 60 days preseason interval;

- improve figure explanations;

- be careful with the use of the term "genetic", acknowledge other sources of variation;

- mention caveats such as masting, beech as extreme case, experimental focus on photoperiod;

- replace population by grid-cell or find other solution for the problem mentioned by the third reviewer;

- explain more clearly the nature of extracted data (site- or individual-level?) and the statistical analysis used for each data set;

- consider the additional references mentioned below.

Separate reviews (please respond to each point):

*Reviewer #1:*

This paper shows that increased pre-growing-season temperatures in central Europe lead to a wider spread of leaf-out and flowering-start days among individuals in several tree and other plant species. The potential causes – in particular within-population variation in preseason-temperature, photoperiod and chilling degree-day requirements – are analyzed. In addition, the slower increase in temperature in spring with increasing overall temperatures plays a role in the reduced synchrony of phenological events within species.

The authors discuss potential consequences, i.e. reduced genetic exchange between individuals due to different flowering times and altered interactions with herbivores due to different leaf-out times. The increased within-population variation will also reduce the between-population variation across sites. Additionally, but not discussed by the authors, it may reduce between-species variation within sites, which could affect niche separation between species and thus potential biodiversity effects on pollinators and herbivores.

Generally, this paper is very comprehensive, as it uses different sources of evidence, including experiments, and careful analyses of data. It describes previously unknown phenomena for a representative set of plant species and analyzes underlying causes. Thus, I have only two major points that in my view need clarification:

1) In several places the authors refer to synchrony between species to derive or justify that here they looked at synchrony within populations. However, I think the logic is not as sound as it could be. In particular, if they do want to make a link between the two levels, they should rather discuss the potential anti-correlation mentioned above, i.e. that a loss of phenological "specialization" (synchrony) of species will lead to more overlap (synchrony) between species at community level. I understand that the authors did not directly analyze the latter because their species were not selected randomly within pixels, but at least a discussion would be useful (or they might even have some data to do the analysis, because for that species with less complete data could also be included).

2) An important assumption in the analysis of effects of pre-season temperature is that the time of 60 days before leaf unfolding or flowering start is the correct one. It would be important to know if other time intervals would have led to similar results, as currently it looks quite arbitrary why 60 days should be best. Such an analysis of different pre-season intervals could be added to the supplement.

The following minor comments are given:

Abstract: delete "genetic".

Delete "Such species-specific responses imply variation in heritable phenological strategies among individuals,".

Introduction, first paragraph: "foliovores"?

Introduction, second paragraph: "common-garden".

Results and Discussion, third paragraph: delete "Surprisingly,".

Results and Discussion, third and fourth paragraphs: "increased".

Results and Discussion, fourth paragraph: LOS and FLS could be analyzed together as in multivariate anova. Just put the two data tables vertically together and add a column "trait".

Results and Discussion, twelfth paragraph: a further useful citation here would be Fakheran et al., 2010.

Results and Discussion, last paragraph: here you may refer to other papers showing "biotic homogenization" due to global change.

Results and Discussion, last paragraph: delete "massive".

Materials and methods subsection “Analysis”, fourth paragraph: delete "and/".

Additional data files and statistical comments:

The statistical analysis is sound. Species identity was used as random term – specific species responses could have been analyzed in more detail with species identity or contrasts thereof as fixed term(s).

*Reviewer #2:*

The effect of temperature change on phenology has attracted a lot of attentions, and many studies reported the change in the timing of plant phenological events. The authors focused on the variation in a population, or synchrony of leaf-out and flowering. Curiously, they found that plant synchrony is low in years with high temperature in temperate forests. They combined evidence of a long-term survey, transplantation and manipulation experiments. The manuscript addressed important questions and reported novel findings.

I have a few comments on the way of presentation of the figures.

First, the figure numbers are often not ordered in number. Please check throughout the manuscript. For example, in the Results and Discussion, Figure 1—figure supplements 4 and 5 appeared in the fourth paragraph, then Figure 2—figure supplement 1, Figure 2—figure supplement 2B, Figure 2—figure supplement 2CFigure in the sixth paragraph, nothing on Figure 2—figure supplement 2A. Figure 4—figure supplement 1 was mentioned at the last sentence of the Materials and methods, but it should be mentioned as Results. The legend of Figure 1—figure supplements 1 and 2 said "Extended Data Figure 5", which does not exist. Figure 1C and D appeared before 1A and B, which may depend on the style of the journal. These are likely remnants of a previous version of the manuscript.

More importantly, how to interpret figures are often not well explained in the main text or figure legends. If a reader reads carefully throughout the manuscript, one may find the logics, but they should be clarified in the main text. "We also simulated synchrony of spring phenology based on the Munich Jan-May temperatures over the past 60 years, assuming that phenology is solely driven by degree-day accumulation (no effect of photoperiod or winter chilling; see Figure 2—figure supplement 2B) and this simulation revealed small losses of synchrony (R2 values between 0.04 and 0.11 and regression coefficients between 0.15 and 0.43, see Figure 2—figure supplement 2C)."

It is not self-evident how Figure 2—figure supplement 2B and C supported these conclusions. To conclude "small", what did you compare with?

Figures"(i) phenological variation among individualsstrongly decreases under short day conditions (Figures 2B and 3B))." It is not self-evident how these figures supported this conclusion. Figure 2B does not show short day directly (photoperiod is there). In Figure 3B, at a first glance, the variation appears high at short day (8-hour day length).

Results and Discussion, ninth paragraph. I would suggest to remove the term "genetic" differences among individuals. I agree that the data suggest most likely, but other possibilities cannot be excluded. For example, plant age or size may affect the temperature responses.

Manipulation and transplantation experiments

Describe more about the nature of the used trees of the manipulation experiments as well as of transplantation experiments. Are they more or less natural trees or from the same region? Genetic data would be the best if any, but historical description would be valuable. In a botanical garden, some of the trees may be transplanted from distant populations. Non-local plants may react to environmental changes differently. Even so, the data supports the individual differences, but it may not support the increase of variation in a small geographic scale.

Some tree species may show masting and may affect the analysis, but it was not clearly mentioned in the text. For masting species, was there no flowering in some years, or did they still flower every year to some extent? Particularly, there are number of studies on the meteorological factors and nutritional effect on the masting of *Fagus sylvatica* and *F. crenata*, two of the species studied in this manuscript (for example, Overgaard et al., 2007 Effects of weather conditions on mast year frequency in beech (*Fagus sylvatica* L.) in Sweden. Forestry, 80, 555-565; Miyazaki et al. 2014 Nitrogen as a key regulator of flowering in *Fagus crenata*: understanding the physiological mechanism of masting by gene expression analysis. Ecol Lett 17, 1299-1309.)

Minor Comments:

Results and Discussion, sixth paragraph: define forcing sums clearly.

Results and Discussion, ninth paragraph: it was not clear if this is a hypothesis or results. By reading later sentences, this seems a hypothesis. Please clarify it.

Legend of Figure 2—figure supplement 1: B is bold but C is not.

*Reviewer #3:*

"Loss of leafout and flowering synchrony under global warming" is an interesting paper on an important topic. The authors have combined long-term data from PEP725 (an impressive database, used for decades in the field) with observations from an arboretum and experiments on *Fagus sylvatica* (henceforth European beech) to attempt to measure changes with climate change in 'within-population' synchrony of leafout and flowering for a handful of species, and then attribute the changes to primarily genetic differences across individuals in photoperiod responses.

Many papers have previously highlighted the potentially increasing importance of genetic variation in phenology with climate change altering the environmental playing field. The paper offers two (I believe new) experiments on European beech, which are an important contribution to the field of phenology.

Unfortunately the paper also suffers from several major flaws that prevent it from having the impact I think it could. I outline these below in no particular order:

1) The authors choose to do experiments on European beech to test their hypothesis that individual variation in predominantly photoperiod responses lead to observed variation in synchrony. Decades of research have highlighted that European beech has the most extreme responses to photoperiod of any species studied, which the authors do not clearly acknowledge.

2) The authors' experiments are ideally designed to test for photoperiod differences. This is because photoperiod is the only factor they fully vary in their experiments (they partially manipulate chilling by taking cuttings across time and their forcing temperatures are always ambient). This means the authors have designed their experiment to maximize differences observed in photoperiod responses, and minimized the potential to find forcing (or chilling) differences.

(*) Pts 1 and 2 taken together mean the most novel parts of the paper were predisposed to find *very* strong photoperiod effects, and this reality should temper any strong conclusions regarding photoperiod in relation to other cues that are drawn from this paper. I think the paper would be stronger if the authors would acknowledge this more clearly.

3) The authors claim to make conclusions about 'populations' but they don't fully define a population. It is defined for PEP725 data and in this case it's a one-degree gridcell. They toss out some PEP725 data based on deviation in altitude or data deemed too variable but I doubt this helps them achieve data that is functionally acting as population. This isn't a death-chime for the paper but I suggest the authors not refer to 'within-population' when what they really mean is 'within-gridcell.' (I'd be comforted to know if their conclusions are altered by the amount of PEP725 data that they removed.)

Relatedly the authors say they use individuals observed over time in the PEP725 data. In my experience with the PEP725 data it is an impressive mishmash of observed crops, planted trees, common garden (e.g., IPG) plants, some herbs and trees etc. in more wild populations etc.. I am not actually sure how one would pull out data from PEP725 in such a way that they know they have the same individual over time. Could the authors clarify this? Or, I think it is potentially more likely the authors have data on the *same site* and *same species* over years but they do not know if it is the same individual. If this is the case the authors should adjust their language.

4) The statistical approaches feel a bit all over the place with no coherent reasoning for what is applied when. We start the paper with a lot of 'XX% of sites did this and XX% were significant.' This is just the sort of data that hierarchical models are designed for – we would then get an overall, estimated-across-site (including uncertainty) estimate – but the authors choose not to use a hierarchical approach. Except they do! In some of the figure captions they seem to use a hierarchical approach for the exact same data and (though I am not sure) I think they sometimes reported 95% confidence intervals in the text and sometimes report 95% credible intervals for the same sort of analysis on the same data.

The authors assert the power and importance of Bayesian hierarchical models in the Materials and methods, thus I suggest they use them throughout. I personally am not much swayed by hearing that '75% of analysed pixels [did X]' and that '18% [were] statistically significant.' There's a whole host of issues with this approach, which the authors seem aware of when the switch to and extol the virtues of hierarchical models later in the paper.

5) The authors use SD to measure synchrony. This should be fine, but it is well known to be biased based on sample sizes. It would strengthen the paper and its conclusions if the authors can show the same results with standard error or some other methods that incorporates bias in sample size.

6) The authors do a good job to review fundamental climatic changes in the shape of spring with climate change, which could drive their results, but they dismiss them without persuading this reader that they don't explain a substantial amount of their variation. They acknowledge they see 'small losses of synchrony' and then reference 'regression coefficients of 0.15-0.43'..…I am not sure what I should compare these numbers to as it's hard to follow, but the other numbers I think I should compare them would be 0.15-0.35 (coefficients given in the preceding paragraphs about changing synchrony) so to me – these numbers seem *very similar*. Could the authors better guide me as to why these regression coefficients should be so easily dismissed? Unless they can show these really are minuscule numbers I suggest they re-think their discussion, which is focused strongly on genetic variation and daylength responses (to compare: the regression coefficients given for heritable differences are 0.09-0.23, are these so different from 0.14-0.43?).

They could, for example, cut what feels like a rather extended, review-paper-style detour into one paper on elevational effects to make more room for a nuanced discussion of the multiple factors that could drive their synchrony findings.

7) I imagine the authors are limited in the references they can include but they could do a better job of showing they are aware of the decades of research in ecology on co-flowering (e.g. CaraDonna, Iler, and Inouye, 2014, or similar work from RMBL) or the evolutionary literature on this, which is quite deep. I think it would also help the authors better interpret their results and better situate their findings in this very large literature, which exists in large part outside the ecological and climate change literature that they focus on.

8) The authors should acknowledge that their arboretum data include variation due to genotype and microclimate as well as a suite of things related to small-site differences, maternal effects etc.. Again, a citation to the common garden literature (which shows how/why multiple gardens are needed) would help.

Minor Comments:

a) The authors seem to be using the extremes of their data when they state 'by up to 55%' (and related day estimates). It's also iffy to work with extremes so it would be helpful for the authors to also provide a mean or median here to better put their results on solid statistical grounds.

b) The statistical methods are generally hard to follow. The authors should write all equations they used (they used random effects, on what? The intercept or slope? Or both? These are critical methods), and they should list their R-hat values and n effective sizes so readers can better evaluate their models.

c) The light values in the experiments are quite low. Is there a reason this was selected? Even for low-light plants like Arabidopsis I think 300 micro mols is more standard.

Additional data files and statistical comments:

See above pts 4-5 and a-b for concerns over statistical approaches.

---

## [Author Response]

Major concerns:These are pointed out in the reviews below. In particular, please:- discuss the issue of within- vs. between-species asynchrony/synchrony;

Done. See subsection “Statistical analyses”, second paragraph.

- justify the choice of 60 days preseason interval;

This is done in the third paragraph of the subsection “Statistical analyses”. See also reply to reviewer #1 point 2.

- improve figure explanations;

Done. See Materials and methods and legend of Figure 2.

- be careful with the use of the term "genetic", acknowledge other sources of variation;

We have deleted ‘genetic’ throughout.

- mention caveats such as masting, beech as extreme case, experimental focus on photoperiod;

See our sixth response to reviewer #2, and our first three responses to reviewer #3. See Results and Discussion, thirteenth paragraph.

- replace population by grid-cell or find other solution for the problem mentioned by the third reviewer;

Done.

- explain more clearly the nature of extracted data (site- or individual-level?) and the statistical analysis used for each data set;

Done. See subsection “Statistical analyses”.

- consider the additional references mentioned below.

Done. See new references CaraDonna, Iler, and Inouye, 2014; Forrest, Inouye and Thomson, 2010 and Fakheran et al., 2010.

Separate reviews (please respond to each point):

Reviewer #1:

[…] Generally, this paper is very comprehensive, as it uses different sources of evidence, including experiments, and careful analyses of data. It describes previously unknown phenomena for a representative set of plant species and analyzes underlying causes. Thus, I have only two major points that in my view need clarification:1) In several places the authors refer to synchrony between species to derive or justify that here they looked at synchrony within populations. However, I think the logic is not as sound as it could be. In particular, if they do want to make a link between the two levels, they should rather discuss the potential anti-correlation mentioned above, i.e. that a loss of phenological "specialization" (synchrony) of species will lead to more overlap (synchrony) between species at community level. I understand that the authors did not directly analyze the latter because their species were not selected randomly within pixels, but at least a discussion would be useful (or they might even have some data to do the analysis, because for that species with less complete data could also be included).

We now discuss that decreasing synchrony within populations should lead to increasing synchrony among species within communities. See Results and Discussion, thirteenth paragraph.

2) An important assumption in the analysis of effects of pre-season temperature is that the time of 60 days before leaf unfolding or flowering start is the correct one. It would be important to know if other time intervals would have led to similar results, as currently it looks quite arbitrary why 60 days should be best. Such an analysis of different pre-season intervals could be added to the supplement.

In their Extended data Figure 2, Fu et al., 2015, show that, for most European tree species, 60 days is the “best” preseason period. The choice of preseason temperature (usually something between 1–3 months) does not affect our results. Ultimately, it is not preseason temperature per sethat affects synchrony, but changes in day-length and/or chilling accumulation, which is why we additionally analyzed the in situ data in this regard (Figure 2) and designed the experiments (Figure 3 and 4).

The following minor comments are given:Abstract: delete "genetic".

Done (see also our response to the fourth major concern above).

Delete "Such species-specific responses imply variation in heritable phenological strategies among individuals,".

Done.

Introduction, first paragraph: "foliovores"?

Herbivores.

Introduction, second paragraph: "common-garden".

Changed.

Results and Discussion, third paragraph: delete "Surprisingly,".

Done.

Results and Discussion, third and fourth paragraphs: "increased".

Changed.

Results and Discussion, fourth paragraph: LOS and FLS could be analyzed together as in multivariate anova. Just put the two data tables vertically together and add a column "trait".

This paragraph describes the correlation between the effect of preseason temperature on a species’ average flowering date and the effect of preseason temperature on flowering synchrony. Leaf-out synchrony (LOS) is not considered here. We agree that LOS and FLS could have been analyzed together, but this would not advance the findings of this study.

Results and Discussion, twelfth paragraph: a further useful citation here would be Fakheran et al., 2010.

Thank you. Now cited.

Results and Discussion, last paragraph: here you may refer to other papers showing "biotic homogenization" due to global change.

Biotic homogenization mainly refers to species invasions and extinctions, a topic that is not covered in this study.

Results and Discussion, last paragraph: delete "massive".

Done.

Materials and methods subsection “Analysis”, fourth paragraph: delete "and/".

Done.

Additional data files and statistical comments:The statistical analysis is sound. Species identity was used as random term – specific species responses could have been analyzed in more detail with species identity or contrasts thereof as fixed term(s).

Reviewer #2:

The effect of temperature change on phenology has attracted a lot of attentions, and many studies reported the change in the timing of plant phenological events. The authors focused on the variation in a population, or synchrony of leaf-out and flowering. Curiously, they found that plant synchrony is low in years with high temperature in temperate forests. They combined evidence of a long-term survey, transplantation and manipulation experiments. The manuscript addressed important questions and reported novel findings.I have a few comments on the way of presentation of the figures.First, the figure numbers are often not ordered in number. Please check throughout the manuscript. For example, in the Results and Discussion, Figure 1—figure supplements 4 and 5 appeared in the fourth paragraph, then Figure 2—figure supplement 1, Figure 2—figure supplement 2B, Figure 2—figure supplement 2C in the sixth paragraph, nothing on Figure 2—figure supplement 2A. Figure 4—figure supplement 1 was mentioned at the last sentence of the Materials and methods, but it should be mentioned as Results. The legend of Figure 1—figure supplements 1 and 2 said "Extended Data Figure 5", which does not exist. Figure 1C and D appeared before 1A and B, which may depend on the style of the journal. These are likely remnants of a previous version of the manuscript.

Figure numbers are now ordered correctly throughout the text, Figure 4—figure supplement 1 is now mentioned in the Results and Discussion section (ninth paragraph).

More importantly, how to interpret figures are often not well explained in the main text or figure legends. If a reader reads carefully throughout the manuscript, one may find the logics, but they should be clarified in the main text. "We also simulated synchrony of spring phenology based on the Munich Jan-May temperatures over the past 60 years, assuming that phenology is solely driven by degree-day accumulation (no effect of photoperiod or winter chilling; see Figure 2—figure supplement 2B) and this simulation revealed small losses of synchrony (R2 values between 0.04 and 0.11 and regression coefficients between 0.15 and 0.43, see Figure 2—figure supplement 2C)." It is not self-evident how Figure 2—figure supplement 2B and C supported these conclusions. To conclude "small", what did you compare with?

We now explain that the regression coefficients obtained by the simulation need to be compared to the regression coefficients in Figure 1—figure supplements 1 and 2 (see Results and Discussion, sixth paragraph).

"(i) phenological variation among individualsstrongly decreases under short day conditions (Figures 2B and 3B))." It is not self-evident how these figures supported this conclusion. Figure 2B does not show short day directly (photoperiod is there). In Figure 3B, at a first glance, the variation appears high at short day (8-hour day length).

Thank you for detecting this error. It should read "(i) phenological variation among individuals strongly increases under short day conditions (Figures 2B and 3B))."

Results and Discussion, ninth paragraph. I would suggest to remove the term "genetic" differences among individuals. I agree that the data suggest most likely, but other possibilities cannot be excluded. For example, plant age or size may affect the temperature responses.

Done (see also our response to the fourth major concern above and our first response to reviewer #1’s minor comments).

Manipulation and transplantation experimentsDescribe more about the nature of the used trees of the manipulation experiments as well as of transplantation experiments. Are they more or less natural trees or from the same region? Genetic data would be the best if any, but historical description would be valuable. In a botanical garden, some of the trees may be transplanted from distant populations. Non-local plants may react to environmental changes differently. Even so, the data supports the individual differences, but it may not support the increase of variation in a small geographic scale.

The in situdata from the PEP database are based on natural trees and we used pixel-level analyses to look at small geographic scales; transplantation experiments were not conducted. For the manipulation experiment, we used individuals grown in the vicinity of the Munich Botanical Garden, which thus should be more or less natural trees from the same region.

Some tree species may show masting and may affect the analysis, but it was not clearly mentioned in the text. For masting species, was there no flowering in some years, or did they still flower every year to some extent? Particularly, there are number of studies on the meteorological factors and nutritional effect on the masting of Fagus sylvatica and F. crenata, two of the species studied in this manuscript (for example, Overgaard et al., 2007 Effects of weather conditions on mast year frequency in beech (Fagus sylvatica L.) in Sweden. Forestry, 80, 555-565; Miyazaki et al. 2014 Nitrogen as a key regulator of flowering in Fagus crenata: understanding the physiological mechanism of masting by gene expression analysis. Ecol Lett 17, 1299-1309.)

For *Fagus sylvatica*, we studied leaf-out behavior, not flowering. We do not know of hard masting behavior in the species for which we had data on flowering phenology, all species flowered (to some extent) in every year.

Minor Comments:Results and Discussion, sixth paragraph: define forcing sums clearly.

Done.

Results and Discussion, ninth paragraph: it was not clear if this is a hypothesis or results. By reading later sentences, this seems a hypothesis. Please clarify it.

Yes, it is a hypothesis. Now clarified.

Legend of Figure 2—figure supplement 1: B is bold but C is not.

Changed.

Reviewer #3:

"Loss of leafout and flowering synchrony under global warming" is an interesting paper on an important topic. The authors have combined long-term data from PEP725 (an impressive database, used for decades in the field) with observations from an arboretum and experiments on Fagus sylvatica (henceforth European beech) to attempt to measure changes with climate change in 'within-population' synchrony of leafout and flowering for a handful of species, and then attribute the changes to primarily genetic differences across individuals in photoperiod responses.Many papers have previously highlighted the potentially increasing importance of genetic variation in phenology with climate change altering the environmental playing field. The paper offers two (I believe new) experiments on European beech, which are an important contribution to the field of phenology.Unfortunately the paper also suffers from several major flaws that prevent it from having the impact I think it could. I outline these below in no particular order:1) The authors choose to do experiments on European beech to test their hypothesis that individual variation in predominantly photoperiod responses lead to observed variation in synchrony. Decades of research have highlighted that European beech has the most extreme responses to photoperiod of any species studied, which the authors do not clearly acknowledge.

We did stress that European beech is unusually photoperiod sensitive, writing "This insight explains why, especially in *Fagus sylvatica*, in which day length has the most pronounced effect on spring phenology(Laube et al., 2014; Zohner et al., 2016), LOS is strongly affected by preseason temperatures (Figure 1C). By contrast, in day-length insensitive species, such as silver birch *Betula pendula* and Norway spruce *Picea abies* (Zohner et al., 2016), preseason warming has a smaller (but still significant) effect on LOS, suggesting that heritable differences in day-length sensitivity are a major driver of within-population phenological variation.”

2) The authors' experiments are ideally designed to test for photoperiod differences. This is because photoperiod is the only factor they fully vary in their experiments (they partially manipulate chilling by taking cuttings across time and their forcing temperatures are always ambient). This means the authors have designed their experiment to maximize differences observed in photoperiod responses, and minimized the potential to find forcing (or chilling) differences.

Individual forcing requirements per secan explain why leaf-out/flowering dates of individuals differ from each other but at least not fully why those differences increase with climate warming. Two approaches were used to show this: i) a simulation based on degree-day models (see our second response to reviewer #2) and even more important ii) an analysis of the standard deviation of degree-days until leaf-out among individuals in relation to preseason temperature (Figures2 and Figure 2—figure supplement 1). The latter analysis shows that it is not only the leaf-out/flowering dates that show greater variation under warmer preseasons but also the required forcing sums. This can only be explained by individual differences in chilling or day-length sensitivity. Forcing requirements of a given individual are never fixed, they always depend on winter chilling and day-length (e.g., Laube et al., 2014).

(*) Pts 1 and 2 taken together mean the most novel parts of the paper were predisposed to find very strong photoperiod effects, and this reality should temper any strong conclusions regarding photoperiod in relation to other cues that are drawn from this paper. I think the paper would be stronger if the authors would acknowledge this more clearly.

Nowhere do we say that it is only day-length that causes losses in synchrony. We state that chilling requirements might play a role (Results and Discussion, tenth paragraph) and we also acknowledge that a flattening temperature curve in spring contributes to synchrony losses (but cannot explain all of the variation we see) [Results and Discussion, sixth paragraph].

3) The authors claim to make conclusions about 'populations' but they don't fully define a population. It is defined for PEP725 data and in this case it's a one-degree gridcell. They toss out some PEP725 data based on deviation in altitude or data deemed too variable but I doubt this helps them achieve data that is functionally acting as population. This isn't a death-chime for the paper but I suggest the authors not refer to 'within-population' when what they really mean is 'within-gridcell.' (I'd be comforted to know if their conclusions are altered by the amount of PEP725 data that they removed.)

We now use “within-pixel” throughout the text. The removal of data is a necessary step to get rid of erroneous entries (e.g., there are leaf-out observations postdating July in the PEP database). In elevational heterogenous pixels, individuals are likely to experience vastly different climate conditions, which is why we removed individuals for which the altitudinal location deviated by >200 m from the average altitude of all individuals within the pixel. See previous comments from a reviewing editor:

Editor: “Due to a low number of individuals per pixel, the authors use relatively large pixels. Did they use corrections for different altitudes within pixels? Or did all individuals within pixels grow at similar altitudes? The authors should provide information about the range of altitudes of individuals within pixels.”

To which we previously replied: “All individuals within pixels grew at similar altitudes (if an individual’s altitude deviated by >200 m from the average altitude of the individuals within the pixel, that individual was excluded from the analysis). This is now stated in the Materials and methods. Below is a histogram of the range of altitudes of individuals within pixels (the average range per pixel was 191 m; the maximum allowed range per definition was 400 m). Our finding that among-individual variation in thermal requirements (degree-days) until leaf-out is increasing under warmer temperatures is unaffected by altitudinal variation because climate data was used for each location separately.”

Relatedly the authors say they use individuals observed over time in the PEP725 data. In my experience with the PEP725 data it is an impressive mishmash of observed crops, planted trees, common garden (e.g., IPG) plants, some herbs and trees etc. in more wild populations etc.. I am not actually sure how one would pull out data from PEP725 in such a way that they know they have the same individual over time. Could the authors clarify this? Or, I think it is potentially more likely the authors have data on the same site and same species over years but they do not know if it is the same individual. If this is the case the authors should adjust their language.

For most tree species, the same individual tree is observed for many years. However, this is not necessarily the case for all time-series, and we now state this in the text (subsection “Data sets”).

4) The statistical approaches feel a bit all over the place with no coherent reasoning for what is applied when. We start the paper with a lot of 'XX% of sites did this and XX% were significant.' This is just the sort of data that hierarchical models are designed for – we would then get an overall, estimated-across-site (including uncertainty) estimate – but the authors choose not to use a hierarchical approach. Except they do! In some of the figure captions they seem to use a hierarchical approach for the exact same data and (though I am not sure) I think they sometimes reported 95% confidence intervals in the text and sometimes report 95% credible intervals for the same sort of analysis on the same data.The authors assert the power and importance of Bayesian hierarchical models in the Materials and methods, thus I suggest they use them throughout. I personally am not much swayed by hearing that '75% of analysed pixels [did X]' and that '18% [were] statistically significant.' There's a whole host of issues with this approach, which the authors seem aware of when the switch to and extol the virtues of hierarchical models later in the paper.

All analyses for which we choose to report percentages of statistically significant correlations are also tested in the light of a Bayesian hierarchical model (Figure 2). We use both simple regression methods as well as Bayesian hierarchical models as we believe that the presentation of the Bayesian models is not enough to make the results understandable and accessible to a broad readership.

5) The authors use SD to measure synchrony. This should be fine, but it is well known to be biased based on sample sizes. It would strengthen the paper and its conclusions if the authors can show the same results with standard error or some other methods that incorporates bias in sample size.

Sample sizes were kept constant across years.

6) The authors do a good job to review fundamental climatic changes in the shape of spring with climate change, which could drive their results, but they dismiss them without persuading this reader that they don't explain a substantial amount of their variation. They acknowledge they see 'small losses of synchrony' and then reference 'regression coefficients of 0.15-0.43'. I am not sure what I should compare these numbers to as it's hard to follow, but the other numbers I think I should compare them would be 0.15-0.35 (coefficients given in the preceding paragraphs about changing synchrony) so to me – these numbers seem very similar. Could the authors better guide me as to why these regression coefficients should be so easily dismissed? Unless they can show these really are minuscule numbers I suggest they re-think their discussion, which is focused strongly on genetic variation and daylength responses (to compare: the regression coefficients given for heritable differences are 0.09-0.23, are these so different from 0.14-0.43?).They could, for example, cut what feels like a rather extended, review-paper-style detour into one paper on elevational effects to make more room for a nuanced discussion of the multiple factors that could drive their synchrony findings.

The estimated regression coefficients are 0.61 for *Fagus sylvatica* and 0.91 for *Alnus glutinosa* (Figure 1—figure supplements 1A and 2A), which is significantly larger than our estimated range of 0.15–0.43 (Figures 1B and Figure 1—figure supplement 1). This is now clarified in the text. To dismiss a flattening temperature curve in spring as the sole driver of reduced synchrony, we also analyze standard deviations in individual degree-day requirements (see also our second responses to reviewer #2 and #3). The regression coefficients of 0.09–0.23 for the effects of chilling on synchrony cannot be compared to those obtained from the degree-day simulation (0.15–0.43) because the response variables are not at the same scale (chill-days and days).

7) I imagine the authors are limited in the references they can include but they could do a better job of showing they are aware of the decades of research in ecology on co-flowering (e.g. CaraDonna, Iler, and Inouye, 2014 or similar work from RMBL) or the evolutionary literature on this, which is quite deep. I think it would also help the authors better interpret their results and better situate their findings in this very large literature, which exists in large part outside the ecological and climate change literature that they focus on.

We now include additional references on co-flowering (CaraDonna, Iler, and Inouye, 2014 and Forrest, Inouye and Thomson, 2010).

8) The authors should acknowledge that their arboretum data include variation due to genotype and microclimate as well as a suite of things related to small-site differences, maternal effects etc.. Again, a citation to the common garden literature (which shows how/why multiple gardens are needed) would help.

We have now deleted the term “genetic” throughout the manuscript. That common-garden data includes variation due to microclimate, etc. is irrelevant to our finding that in warm years variation among individuals is increasing. This can only be explained by individuals responding differently to changing abiotic stimuli.

Minor Comments:a) The authors seem to be using the extremes of their data when they state 'by up to 55%' (and related day estimates). It's also iffy to work with extremes so it would be helpful for the authors to also provide a mean or median here to better put their results on solid statistical grounds.

These percentages are based on the modeled standard deviations among individuals (Figure 1E, F). Instead of using 4 standard deviations (period during which 95% of individuals leaf-out/flower) we could also use only 2 standard deviations, but this does not affect the inferred percentages.

*b) The statistical methods are generally hard to follow. The authors should write all equations they used (they used random effects, on what? The intercept or slope? Or both? These are critical methods), and they should list their R-hat values and n effective sizes so readers can better evaluate their models.*

Equations are now written down in the Materials and methods and the model description has been revised (subsection “Analysis of leaf-out and flowering synchrony (LOS and FLS) using the PEP database”).

c) The light values in the experiments are quite low. Is there a reason this was selected? Even for low-light plants like Arabidopsis I think 300 micro mols is more standard.

To induce day-length responses, light values way below 100 µmols are sufficient.